# Identification of an Embryonic Cell-Specific Region within the Pineapple *SERK1* Promoter

**DOI:** 10.3390/genes10110883

**Published:** 2019-11-01

**Authors:** Aiping Luan, Yehua He, Tao Xie, Chengjie Chen, Qi Mao, Xiaoshuang Wang, Chuhao Li, Yaqi Ding, Wenqiu Lin, Chaoyang Liu, Jingxian Xia, Junhu He

**Affiliations:** 1Tropical Crops Genetic Resources Institute of Chinese Academy of Tropical Agricultural Science, Haikou 571101, China; aipingluan@hotmail.com (A.L.); hejunhu123@163.com (J.H.); 2College of Horticulture, South China Agricultural University, Guangzhou 510642, China; xietao@fosu.edu.cn (T.X.); ccj0410@gmail.com (C.C.); zjmaoqi@163.com (Q.M.); Luckygirlkeepsmile@163.com (X.W.); lichuhao@scau.edu.cn (C.L.); dyqkxy2019@126.com (Y.D.); linwenqiu1989@163.com (W.L.); liuchaoyang@scau.edu.cn (C.L.); xjx0036@scau.edu.cn (J.X.); 3College of Agricultural, Guangdong Ocean University, Zhanjiang 524088, China

**Keywords:** pineapple, *AcSERK1*, embryonic cell-specific promoter, regulatory sequences, somatic embryogenesis

## Abstract

Plant tissue culture methods, such as somatic embryogenesis, are attractive alternatives to traditional breeding methods for plant propagation. However, they often suffer from limited efficiency. Somatic embryogenesis receptor kinase (*SERK*)1 is a marker gene of early somatic embryogenesis in several plants, including pineapple. It can be selectively induced and promotes a key step in somatic embryogenesis. We investigated the embryonic cell-specific transcriptional regulation of *AcSERK1* by constructing a series of vectors carrying the *GUS* (Beta-glucuronidase) reporter gene under the control of different candidate *cis*-regulatory sequences. These vectors were transfected into both embryonic and non-embryonic callus, and three immature embryo stages and the embryonic-specific activity of the promoter fragments was analyzed. We found that the activity of the regulatory sequence of *AcSERK1* lacking −983 nt ~−880 nt, which included the transcription initiation site, was significantly reduced in the embryonic callus of pineapple, accompanied by the loss of embryonic cell-specific promoter activity. Thus, this fragment is an essential functional segment with highly specific promoter activity for embryonic cells, and it is active only from the early stages of somatic embryo development to the globular embryo stage. This study lays the foundation for identifying mechanisms that enhance the efficiency of somatic embryogenesis in pineapple and other plants.

## 1. Introduction

The pineapple (*Ananas comosus* L.) is a perennial herb from the Bromeliaceae family and one of the world’s most important tropical fruit species. Pineapple is largely vegetatively propagated because the species is self-sterile. This has traditionally involved sucker propagation, which is a simple and low-cost process. However, it suffers from several significant limitations, including a low reproductive coefficient, prolonged production periods, and non-uniform growth and development [1]. To avoid these problems, pineapple breeding by tissue culture has been developed as an attractive alternative to traditional breeding methods. This approach has several key advantages, including rapid and uniform growth and development of plants, a two-month reduction in production cycle, and viral disease reduction. 

Organogenesis and somatic embryogenesis are the two in vitro tissue culture techniques used for plant regeneration, and both have been applied to the pineapple [1,2,3,4]. However, unlike organogenesis, somatic embryogenesis (SE) is an ideal rapid propagation method for many different types of in vitro plants and trees and fits the requirements of industrialization [5]. The transition from somatic cell to embryonic cells is a key stage in SE, and appears to be a critical step limiting efficiency for plant regeneration [6,7,8]. Thus, understanding the mechanisms involved in the somatic to embryonic cell transition in order to improve SE efficiency has become a major focus in the field [9].

The SE pathway in animals and plants is regulated by a variety of factors [10]. Somatic embryogenesis receptor kinases (*SERKs*) are a small gene family of receptor-like kinases that play diverse roles in plants, including in SE, pluripotency, reproductive development, the immune response, and stomatal patterning [11,12]. According to the NCBI database, *SERK* genes have been found, thus far, in 49 species. Most of the *SERK* genes from different species share a similar gene structure [13]. Among the *SERK* family, *SERK1* is regarded as an SE marker gene of pineapple and other plants [14,15]. *SERK1* expression is associated with induction of SE and also promotes the transformation of somatic cells to embryonic cells. For instance, expression of *SERK1* in *Arabidopsis thaliana* was shown to increase its SE capacity by three to four times [16]. *SERK1* expression can be induced by plant hormones, such as 6-BA, NAA [17], and 2,4-D [18], and by disease defense signaling molecules, such as SA, BTH, JA, and ABA [14,19]. This would suggest that its regulatory sequences/promoter contains a corresponding *cis*-acting element that responds to the induction factors [20,21,22,23,24,25]. While the *SERK1* gene shows enhanced expression in embryogenic cells in many other plant species upon treatment with plant growth regulators, it is normally also expressed in a subset of somatic cells. In *Arabidopsis thaliana*, *AtSERK1* is first expressed in the developing ovule [16]. In maize, *ZmSERK1* is preferentially expressed in male and female reproductive tissues and is most strongly expressed in microspores [26]. In rice, *OsSERK1* promote the differentiation of rice callus into adventitious buds [19] and is likely to play a more prominent role in non-embryonic tissues [27]. The expression of *SERK* during somatic embryogenesis of potato revealed that it increased during induction. There was no change in *StSERK* gene expression during subsequent embryonic development and embryo maturation, and *StSERK* had different levels of expression in other plant organs (leaves, seeds, tubers, and flower buds) [28]. These findings indicate that *SERK1* predicts embryogenic potential but is not exclusively associated with early embryogenesis. 

In our previous study, we cloned three *SERK* genes from pineapple [14,29]. Subsequent bioinformatics analysis indicated that they belonged to the leucine-rich repeat receptor-like kinase (LRR-RLK) gene family and possessed the characteristic conserved domain and the conserved exon / intron structure shared by the *SERK* gene family. Only very low expression levels of the three *AcSERKs* could be detected in non-embryonic callus, roots, stems, leaves, calyx, bracts, petals, anthers, ovary and ovules, parenchyma cells in young stems, parenchyma cells in the stem cortex, meristematic cells in roots, and unexpanded young leaves. Expression of all three *AcSERKs* could be induced by mechanical injury, salicylic acid, methyl jasmonate, and high salt treatment. In addition, unlike *AcSERK3*, expression of *AcSERK1* and *AcSERK2* could also be induced by low temperature. The hybrid signal of *AcSERK1* appeared strongly in single embryonic cells, then gradually weakened, remaining detectable till the early globular stage of somatic embryos, and then further decreased in the late globular stage [14,29]. According to *AcSERK1* expression characteristics, an embryonic cell-specific region was predicted to be present in its regulatory sequence.

In order to study the transcriptional regulation mechanism of *AcSERK1*, we previously cloned the promoter of *AcSERK1*. We found that the TSS (Transcription Start Sites) (+1) was the 258th nucleotide (G) upstream from the ATG (initiation codon), the length of the general promoter region was 2090 bp, and the length of the 5′-UTR was 258 bp (+1~+258). A recombinant vector consisting of the *AcSERK1* 5′-upstream region (2090~+258) and the reporter gene *GUS* was constructed to enable transient transformation into different organs, and the embryonic and non-embryonic callus of pineapple, to analyze its expression patterns. Subsequent histochemical staining showed no obvious staining in the leaves, stems, roots, anthers, petals, ovary, and non-embryonic callus. Notably, *GUS* staining was observed only in 2,4-D-induced embryonic callus. This indicates that the complete regulatory sequence in the 5′-upstream region of *AcSERK1* had no activity in non-embryonic cells and, therefore, that its promoter activity was specific to embryonic cells [30]. In this study, we constructed a series of deletion vectors with *GUS* as a reporter gene, which were constructed to further isolate and characterize the embryonic cell-specific region. 

## 2. Materials and Methods

### 2.1. Vector Construction

Using GUS as a reporter gene, a series of recombinant expression vectors containing the *AcSERK1* upstream regulatory sequences with different deletions (Appendix A) were constructed, with reference to the pAS2 (−2090~+258) :: *GUS* recombinant vector construction procedure [30]. Briefly, this procedure involved replacing the *CaMV* 35S promoter in the binary expression vector pBI121 using the *XbaI* and *HindIII* sites with the complete 5′ upstream regulatory sequence of *AcSERK1* by homologous recombination with the primers AS1-F (TGATTACGCCAAGCTTATAAATAATTAGACACTTCACGCAAC), and AS1-R (CCGGGGATCCTCTAGATGCCGCCGCCGCGAGCT). The position of *cis* elements in the *AcSERK1* upstream regulatory sequences was predicted using the signal scanning function of the PlantCARE database [31]. Subsequently, specific primers were designed to avoid destroying the predicted *cis*-acting elements and various deletion regions were amplified by PCR (primers shown in Appendix A). Primers were designed to introduce 15 bp homologous sequences on both sides of the *XbaI* and *HindIII* cleavage sites at the 5′ and 3′ ends of the promoter deletion region. The vector was linearized with the restriction enzymes *XbaI* and *HindIII*, and construction of the recombinant plasmids was completed by homologous recombination of the linearized vector and the amplified promoter deletion regions using In-Fusion enzyme (TaKaRa, Dalian China). Next, the plasmid was transformed into *Escherichia coli* DH5α strain by heat shock, and positive clones were identified by sequencing. The plasmids extracted from the positive clones were transfected into *Agrobacterium tumefaciens* GV3101 by heat shock. All PCR amplified DNA fragments were verified by sequencing.

### 2.2. Plant Materials

The pineapple species, ‘Shenwan’ (*Ananas comosus* L.), used was collected from the garden of South China Agricultural University in Guangzhou, China, in 2013. Suckers were obtained from the pineapple and cultured in embryonic induction medium (Murashige and Skoog [MS] + 5 mg/L 2,4-D + 0.5 mg/L BA) to induce embryonic callus, or in medium without 2,4-D (MS + 2 mg/L NAA + 3 mg/L BA) for non-embryonic callus [32]. The culture obtained by induction of somatic embryos for 40 days (embryonic callus mainly containing globular embryo) was transferred to somatic embryo development medium (MS + 1 mg/L NAA + 0.5 mg/L BA) for 10 days (mainly developing into pyriform embryo) and 20 days (mainly developing into bamboo shoot embryo). The embryonic and non-embryonic callus and immature embryos (globular embryo, pyriform embryo, and bamboo shoot embryo) were then used for transient transfection experiments (Section 2.3).

### 2.3. Transient Transfection 

Callus was collected after 5, 10, 15, 20, 25, 30, 35, 40, 45, and 50 days of culture in embryonic induction medium, or after culture in non-embryonic induction medium (embryonic callus and non-embryonic callus, respectively) for transient transformation. Callus for observation of promoter activity in immature embryos was collected after being cultured for 10 days and 20 days in somatic embryo development medium (see above). The collected callus were cut into~3 × 3 × 3 mm cubes for immediate Agrobacterium-mediated transient transformation. Briefly, the transfected *Agrobacterium tumefaciens* GV3101 was cultured by streaking, and single colonies were picked and inoculated into a YEB (Yeast Mannitol Medium) liquid medium and cultured overnight at 28 °C with shaking. Once the concentration reached an absorbance value of ~0.5 at 600 nm (OD_600_ = 0.5), the Agrobacterium solution was used for the transient transformation experiments. In order to eliminate the error caused by different conversion efficiencies and protein extraction efficiencies between different batches, an internal standard vector (*CaMV* 35S :: Fluc) was introduced with the test vector [33]. Note that the test vectors and the internal standard vectors were prepared separately. The final mixture was prepared with a ratio of test carrier: internal standard carrier = 2:1 (*v:v*). Next, the pineapple callus samples were infected with this mixture using vacuum infiltration [34]. After infection, the callus samples were cultured in the dark on medium containing MS + 2 mg/L NAA + 3 mg/L BA for two days. Subsequently, 10 pieces of callus from each material transformed with a different deletion expression vector were collected for GUS histochemical staining. In addition, ~1 g of each material was immediately fixed with liquid nitrogen and stored at −80 °C for quantitative assays (Section 2.4).

### 2.4. GUS Histochemical Staining

Ten pieces of callus for GUS histochemical staining were immersed in X-Gluc staining solution (1 mM X-Gluc, 0.1 M sodium phosphate buffer pH 7.0, 0.1% Triton X-100, 8 mM-mercaptoethanol) and incubated at 37 °C in the dark for 16 h. After triplicate decolorization by 75% ethanol at 65 °C for 30 min, samples were analyzed using an Olympus stereo microscope.

### 2.5. Promoter Activity Assay

The quantitative promoter activity assays using GUS fluorescence in embryonic versus non-embryonic callus were performed using methods described previously [33,35]. Briefly, the callus stored at −80 °C was ground with liquid nitrogen, and 100 mg of the sample powder was added to 400 μL of cell culture lysis reagent (CCLR, Promega, Madison, WI, USA). After mixing, the mixture was put in an ice bath for 10 min, and then centrifuged at 13,000× *g* for 5 min. The supernatant was dispensed into a new centrifuge tube for GUS and Fluc (Firefly Luciferase gene, internal control) assays. For the GUS assay, 20 μL supernatant was taken and immediately mixed with 480 μL reaction solution (0.1M sodium phosphate buffer pH 7.0, 0.5M EDTA pH 8.0, 0.1% Triton X-100, 10mM-mercaptoethanol, 0.1% sodium N-laurosylsarcosine, 1 mM 4-Methylumbelliferyl-b-d-glucuronide), and incubated in a 37 °C water bath. One hundred microliters of the mixture was taken at 0 min and 30 min, and 900 μL of 200 mM sodium carbonate buffer was added to each sample to terminate the reaction. After mixing, 200 μL of sample was taken and measured using the Hitachi F-4600 Fluorescence Spectrophotometer (Tokyo, Japan) with the emission light set to 465 nm and the excitation light set to 340 nm. A GUS activity unit is nmol 4-methyl umbelliferone (4-MU) minute^−1^. The detection of the luminous value of Fluc was carried out following the instructions of the Promifer Luciferase Assay System (E1500). Thermo Scientific Luminoskan Ascent (Waltham, MA, USA) was used, with 10 s as the reading time.

The fluorescence values of GUS and the luminescence value of Fluc were detected separately, and the ratio of the two tests was set as the initial value of the promoter activity. Note that the initial value unit was nmol 4-MU min^−1^ per light units/10 s. To determine embryonic cell-specific promoter activity, we used the ratio of the initial value of the promoter activity in the embryonic tissue to that in the non-embryonic tissue. A relative promoter activity of 1 refers to roughly equal promoter activity in both embryonic and non-embryonic callus.

### 2.6. Gene Expression Analysis

Total RNA from each callus sample was extracted using TRIzol reagent (Invitrogen, Shanghai, China) and cDNAs were synthesized using PrimeScript^TM^ RT reagent Kit with gDNA Eraser (TaKaRa, Dalian, China). Quantitative real-time PCR (qPCR) was performed by Thunderbird SYBR qPCR Mix(Toyobo, Shanghai, China)in the iQ5 Real-time PCR system (BioRad, Hercules, CA, USA). The primers used are listed in Table 1. The specificity of primers was confirmed by melting curve analysis. Each reaction was performed in biological triplicates, and the relative gene expression values were calculated using the 2^−△△CT^ method. The expression levels were normalized against the pineapple *β-actin* gene [14]. 

## 3. Results

### 3.1. Analysis of the Promoter Activity of the Complete 5′ Upstream Regulatory Sequence of AcSERK1 during the Induction of Pineapple SE

The transition from somatic cell to embryonic cell is one of the key stages limiting SE efficiency for plant regeneration [6,7,8]. *SERK1* is known to promote SE, and we previously identified promoter activity in the complete 5′ upstream regulatory sequence (−2090~+258) of *AcSERK1* that was specific to embryonic cells [30]. We first sought to further dissect this regulatory sequence and identify the region required for its activity during SE. We generated embryonic callus by culturing suckers in embryonic induction medium (containing MS + 5 mg/L 2,4-D + 0.5 mg/L BA, see methods) for 0 (i.e., non-embryonic callus), 10, 20, 30, 40, and 50 days. The callus was then collected at each timepoint and used for Agrobacterium-mediated transient transformation with a recombinant expression vector containing the previously isolated *AcSERK1* regulatory sequence upstream of the GUS reporter gene (pAS2 [−2090~+258] :: *GUS* reporter), and incubated for two days. In callus cultured for only 10 days in embryonic induction medium, weak GUS histochemical staining was detectable over a very small surface area of the callus (Figure 1A). In contrast, callus collected after 20 days of induction displayed more extensive GUS staining, which increased in area and intensity in callus collected after 30 days of induction. The deepest staining and largest stained area was observed in callus after 40 days of induction (Figure 1A, indicated by arrows), which dramatically decreased after 50 days. As a negative control, we transfected the pAS2 [−2090~+258] :: *GUS* reporter into plants grown from somatic embryos and observed no staining. Thus, *SERK1* promoter activity increases over the course of SE, before subsequently decreasing. 

We next quantified the activity of *AcSERK1* during SE induction by quantifying the expression levels of the *GUS* gene using qPCR analysis. qPCR of the co-transformed *CaMV* 35S promoter upstream of the *Fluc* gene was used as a control for transformation efficiency (see methods). Consistent with the data above, the expression of the pAS2 (−2090~+258) :: *GUS* reporter first increased up to 40 days induction, and then decreased, while *Fluc* expression remained relatively constant (Figure 1B). To measure the embryonic-specific promoter activity of pAS2 (−2090~+258) during SE induction at the protein level, we quantified GUS fluorescence levels in the embryonic vs non-embryonic tissues using spectrophotometry (see methods). A relative promoter activity value of 1 indicates equal promoter activity in both embryonic and non-embryonic callus. Under the same transformation conditions, we found that the relative promoter activity of pAS2 remained fairly constant between 0 and 10 days of induction (Figure 1C), then gradually increased reaching a peak on day 40 at 26 times the levels in non-embryonic tissues. Finally, there was a significant decrease to three times non-embryonic levels. Thus, promoter activity monitored at the expression level followed a similar trend to the *GUS* qPCR assay result. Together, these results further support a role for *AcSERK1* specifically in SE and reveal the dynamics of promoter activity over the course of SE. 

### 3.2. Identification of the Embryonic Cell-Specific Region in the 5′ Upstream Regulatory Region of AcSERK1

Given the strongest activity of the 5′ upstream regulatory sequence of *AcSERK1* was observed in callus after 40 days of SE induction by 2,4-D, we used this material for our deletion analysis. We generated a series of 12 deletion mutants of the promoter, working around predicted *cis* regulatory sequences (see Methods and Appendix A). Data from only a selection of this series needed to pinpoint the active promoter region are shown. In embryonic callus transformed with the deletion mutant pAS4 (−1138~+258) :: *GUS*, and the positive control, *CaMV* 35S :: *GUS*, we observed strong GUS histochemical staining (Figure 2A). However, embryonic callus transformed with pAS5 (−772~+258) :: *GUS* in which the deletion further expanded to 772 nt upstream relative to the TSS was stained only faintly. Thus, the promoter activity is located between −1138 and −772. Further expression vectors were designed to span this region. Almost no staining was observed in embryonic callus after transient transformation with the recombinant expression vector pAS15 carrying a deletion of (−983/−880), whereas intensive staining was observed with pAS16: the (−880/−772) deletion (Figure 2A). Only very faint staining was observed in embryonic callus transformed with the negative control pCK (*GUS*-nos) (Figure 2A).

We next quantified the relative promoter activities as above of the different mutants, which are shown in Figure 2B. When deletion in the regulatory sequence expanded to 772 nt upstream relative to TSS (pAS5), its relative promoter activity was 1 (i.e., equal in both embryonic and non-embryonic callus). In contrast, when the deletion of the regulatory sequence was set only to −1138 nt, the relative activity of pAS4 (−1138~+258) significantly increased, reaching 37.1. The relative promoter activity of the regulatory sequence with deletion of (−880/−772) was 37.0 (pAS16), whereas the middle deletion of (-983/−880) was 1.1 (pAS15). Thus, consistent with the *GUS* staining results, these data indicate that deletion of (−983/−880) directly affects the embryonic cell-specific promoter activity of the *AcSERK1* regulatory sequence.

We further analyzed three immature embryo stages by *GUS* staining. The stages of the immature embryo in pineapple include the globular embryo, pyriform embryo, and bamboo shoot embryo stages, similar to the development of the zygotic embryo. The globular embryo was transformed with the pAS2 (−2090~+258) :: *GUS* reporter, and we observed strong *GUS* histochemical staining (Figure 3A). However, almost no staining was observed in the pyriform embryo or the bamboo shoot embryo after transient transformation with pAS2 (Figure 3B,C). In the three immature embryo stages respectively transformed with the pAS15 vector carrying a deletion of (−983/−880), we also observed almost no *GUS* histochemical staining (Figure 3D,E,F). These *GUS* staining results of the globular embryo stage again confirmed that deletion of (−983/−880) directly affects the embryonic cell-specific promoter activity of the *AcSERK1* regulatory sequence. 

### 3.3. Bioinformatics Analysis

The 103 bp region between −983 nt and −880 nt was predicted by bioinformatics analysis (see methods) to contain only three CAAT-box elements (−CAAAT, +CAAT, and −CAAT, where “+” “−” indicate coding or noncoding chains) (Figure 4). In addition, NCBI blast of the 103 bp region in the *AcSERK1* promoter with the 5′ upstream regulatory sequence of *AcSERK2* and *AcSERK3*, revealed only 11 bp of homologous sequence (sequence information is indicated by arrows in Figure 4) with *AcSERK3*, and no homologous sequence (length >4 bp) with *AcSERK2*. This provides further support of an embryonic-specific function for *AcSERK1*. In green plants, 40 coding genes (Appendix A) associated with somatic embryogenesis from the *SERK* family can be retrieved from the NCBI. However, only promoters of three pineapple *SERK*s could be found which were submitted by our group. Using reciprocal BLAST (BlastP with –evalue 1e^−5^) strategy, we characterized 142 SERK gene family members from 20 species (Appendix A) and obtained their promoter sequences. After a thorough sequence search (BlastN with –evalue 1000 -word_size 7 -gapopen 5 -gapextend 2 -penalty -3 -reward 1), we found that the isolated region is not conserved, but CAAT-box elements are conserved, among the promoters of these genes (Appendix A).

## 4. Discussion

The expression of genes is regulated by the 5′ upstream regulatory sequence and the transcription factors interacting with it. Therefore, studying the functional regions contained in the 5′ upstream regulatory sequence is of importance for gene regulation study [38,39]. Our findings reveal that the promoter activity of the 5′ regulatory sequence of *AcSERK1* gradually increases during the transformation from non-embryonic cells to embryonic cells and then to the formation of globular embryos. In contrast, in the late phase of globular embryo development, promoter activity gradually weakens. This validates the results of our previous study showing the gene expression pattern and promoter activity of *AcSERK1* [14,30].

Our previous studies demonstrated that the 5′ regulatory sequence of *AcSERK1* has embryonic cell-specific promoter activity under non-stress conditions. However, other researchers have also shown that activity of the 5′ regulatory sequence of *AcSERK1* is inducible under dark and low temperature conditions [22]. Thus, identification of the functions of different regions in the 5′ regulatory sequence is essential for studying the corresponding transcriptional regulation and biological function of the protein. Our deletion assay for identifying these *cis*-acting elements in a regulatory sequence, by monitoring expression of deletion vectors containing *GUS* or *GFP* has been widely used [40,41]. The active region required specifically for expression during SE verified here offers an important starting point for the subsequent isolation and identification of embryonic cell-specific elements and the transcription factors interacting with it. Relevant for this, we did identify three predicted CAAT-box elements within the 103 bp sequence of −983 nt~−880 nt. Interestingly, the direction of the embryonic cell-specific region in the *AcSERK1* promoter was opposite to the 11 bp homologous sequence in the 5′ upstream regulatory sequence of *AcSERK3*. Therefore, whether the 11 bp homologous sequence possesses a similar biological function in the 5′ regulatory sequence of *AcSERK1* and *AcSERK3* also needs further study.

So far, five cases of the 5′ upstream regulatory sequence isolated from the *SERK1* gene have been reported. They are the 2000 bp 5′ upstream regulatory sequence of *AtSERK1* isolated from *Arabidopsis thaliana* [42]; the 2.4 kb 5′ upstream regulatory sequence of *OcSERK1* isolated from rice [27]; the 1.5 kb 5′ upstream regulatory sequence of *MtSERK1* isolated from *Medicago truncatula* [17]; and the 2348 bp 5′ upstream regulatory sequence of *AcSERK1* from pineapple in our previous study [30]. These studies isolated the 5′ upstream regulatory sequences of *SERK1* and analyzed its promoter activity in different organs, as well as in somatic embryos, without identification of the different functional regions within them. Compared with the previous studies, our research goes further by validating the cell-specific promoter activity of the *AcSERK1* 5′ upstream regulatory sequence, isolating its functional region by 5′ deletion analysis, and further defining the minimal functional unit of the promoter with a more directed deletion analysis. Although in this study a shorter functional region with embryonic cell-specificity was isolated, the *cis*-acting element in it was not defined. This would require further deletion analysis and point mutations to identify the conserved sequence of the embryonic cell-specific *cis*-acting element. 

## 5. Conclusions

In summary, *SERK1* is a functional gene in the early stage of somatic embryogenesis that likely acts upon the somatic embryogenesis regulatory network. Thus, its upstream transcriptional regulation is of great importance for increasing the rate of somatic embryogenesis. Our isolation and identification of the embryonic cell-specific region within the promoter of *AcSERK1* will help to decipher the molecular mechanisms of somatic embryogenesis and thereby improve the efficiency of pineapple SE and in vitro propagation.

## Figures and Tables

**Figure 1 genes-10-00883-f001:**
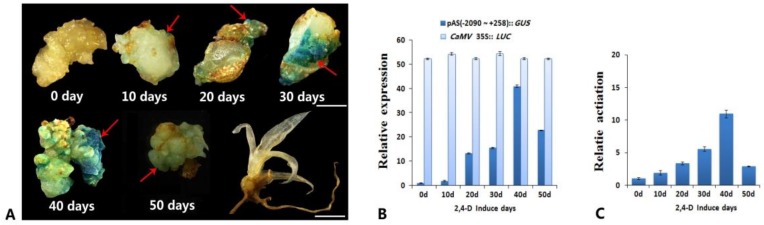
Transient expression analysis of promoter activity of the 5’ upstream regulatory sequence of *AcSERK1*. Callus was isolated after different lengths of time cultured in embryonic induction medium and analyzed by GUS histochemical staining. Red arrows indicate areas of *GUS* expression: (**A**) 0 days (d, i.e., non-embryonic callus), 10 days, 20 days, 30 days, 40 days, 50 days, and plant regenerated from somatic embryo (negative control). Scale bar of callus indicates 2mm, Scale bar of plant regenerated from somatic embryo indicates 5mm. (**B**) Relative expression levels of the *GUS* gene measured by qPCR during the induction of pineapple SE(somatic embryogenesis). Pineapple β-actin gene served as the internal control. Data represent three biological replicates and error bars denote standard error of the mean. (**C**) Quantitative *GUS* assay to determine embryonic specific promoter activity. Data plotted as relative promoter activity: a value of 1 indicates equal promoter activity in both embryonic and non-embryonic callus. Data represent three biological replicates, and the error bars denote standard error of the mean.

**Figure 2 genes-10-00883-f002:**
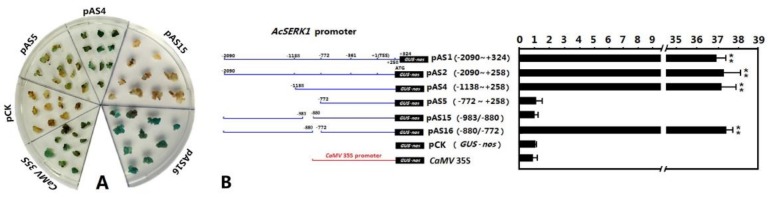
Deletion analysis of the embryonic cell-specific region in the 5′ upstream regulatory region of *AcSERK1*. (**A**) The results of GUS staining in the embryonic callus, (**B**) The quantitative measurements of promoter activity. The numbers in the vector map (left) indicate the deletion position. The vectors were co-infected with the internal standard vector *CaMV* 35S :: Fluc (pBI121-Fluc) into pineapple callus (embryonic and non-embryonic). The bars represent relative promoter activities (the ratio of GUS activity to Fluc activity) in the embryonic callus to those of the non-embryonic callus (i.e., embryo-specific promoter activity). The data represent the average of three biological replicates, and the error bars denote Standard Deviation (SD). ** highly significant (*p* < 0.01), one-tailed *t*-test.

**Figure 3 genes-10-00883-f003:**
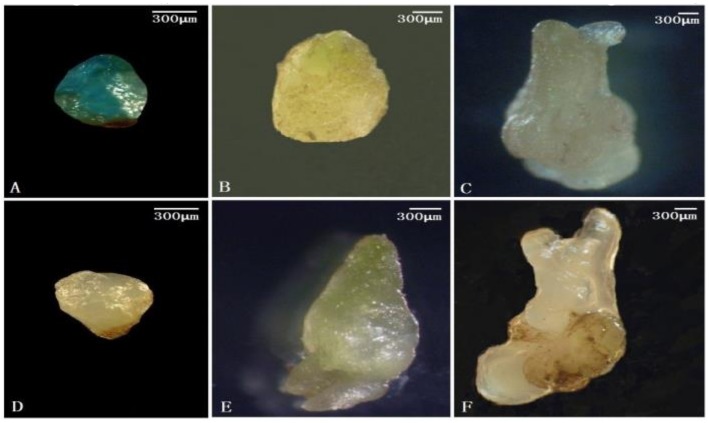
Transient expression analysis of promoter activity with pAS2 (−2090~+258) :: *GUS* and pAS5 (−772~+258)::*GUS* in three stages of immature embryos.

**Figure 4 genes-10-00883-f004:**
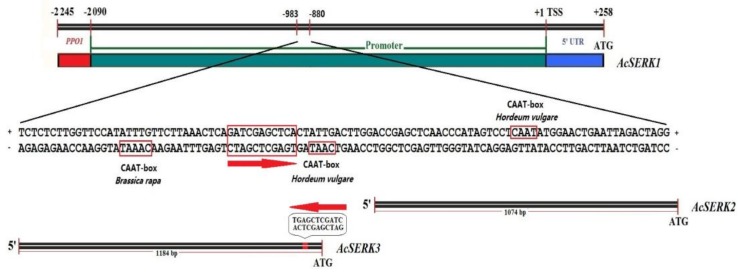
Bioinformatics analysis of embryonic cell-specific sequences.

**Table 1 genes-10-00883-t001:** Primers used for qPCR.

Primer Name	Primer Sequence	References
GUS_F_	5′-AACCGTTCTACTTTACTGGCTTTGG-3′	Wang et al., 2013 [36]
GUS_R_	5′-GCATCTCTTCAGCGTAAGGGTAAT-3′	
Fluc-_F_	5′-TGCACATATCGAGGTGGACATC-3′	Murray et al., 2017 [37]
Fluc-_R_	5′-AGAACCAGAAGAATTTGCAGCAT-3′	
β-actin_F_	5′-CTGGCCTACGTGGCACTTGACTT-3′	Ma et al., 2012 [14]
β-actin_R_	5′-CACTTCTGGGCAGCGGAACCTTT-3′	

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
