# Peer review of "Identification of an Embryonic Cell-Specific Region within the Pineapple SERK1 Promoter"

_genes, 2019, doi:10.3390/genes10110883_

Round 1
Reviewer 1 Report
This is a nice and comprehensive study of the AcSERK1 promoter that identifies a previously unrecognized element of approx 100 bases that confers embryonic expression in pineapple callus. I have a few suggestions that I think should be addressed.
The Introduction is now quite long and may be reduced without clarity Please include a more detailed analysis of the promoters of the 40-odd SERK1 genes that have been published or are available in the database to see how much sequence similarity there is and whether the pattern and spacing of the three CAAT boxes is indeed found in all or only in few. This would help to see if the 10% identity found between the 5 genes that have been analyzed in detail is indeed significant or not. Please also indicate which BLAST settings were used and include approx 2-3 kb of upstream sequences. Reference 41 is wrong for the AtSERK1 promoter. please include the correct reference.Author Response
Point 1: The Introduction is now quite long and may be reduced without clarity.
Response 1: Thanks for your advice. We admit that the “Introduction” part is too long and we have reduced it.
Point 2: Please include a more detailed analysis of the promoters of the 40-odd SERK1 genes that have been published or are available in the database to see how much sequence similarity there is and whether the pattern and spacing of the three CAAT boxes is indeed found in all or only in few. This would help to see if the 10% identity found between the 5 genes that have been analysed in detail is indeed significant or not. Please also indicate which BLAST settings were used and include approx 2-3 kb of upstream sequences.
Response 2: Thanks for the critical comments. After a thorough search in public sequence databases including NCBI, DDBJ and EBI, we could only find promoter sequences of AcSERK1-3, which were submitted by our groups previous. However, as the reviewer suggested, a broader analysis would be helpful. Thus, we have tried our best to obtain potential promoter sequences (2 kb upstream to ATGs) of SERKs from 20 plant species. Using BlastN with sensitive parameters, we found that CAAT-boxes are found in all promoters (Fig S3), which is the same observation as our previous analysis. We have also appended these results in the manuscript. Thanks for your constructive advice again.
Point 3: Reference 41 is wrong for the AtSERK1 promoter. Please include the correct reference.
Response 3: You are right. We have revised “reference 41” to the correct one. Besides, we have also checked all citations to ensure they refer to the correct papers.
Reference #41:Valérie, H.; Jean-Philippe, V.; Marijke, V. H.; Ed, D.L.S.; Kim, B.; Ueli, G.; Sacco, C. V. The Arabidopsis Somatic Embryogenesis Receptor Kinase 1 Gene Is Expressed in Developing Ovules and Embryos and Enhances Embryogenic Competence in Culture. Plant physiology. 2001, 127(3), 803-816.

Reviewer 2 Report
This is a well reasoned and technically sound exploration of the structure of the SERK1 upstream regulatory sequence from pineapple. The result is a clear demonstration of an important cis-regulatory region. The main criticisms I have relate to errors in the citations and unclear wording.
For instance, line 349 cites reference #41 to refer to a study of the Arabidopsis SERK1 upstream regulatory sequence, but the citation is Becker et al 1992, describing binary vectors. Please check all citations to ensure they refer to the correct papers.
A bigger issue is line 64 where the authors cite reference #14 to claim they have shown over-expressing SERK1 in pineapple doubles the efficiency of SE. However, that paper doesn't make that claim, rather it only describes the expression domain and dynamics of SERK1. Later (line 323) the authors again mention doubling SE in pineapple, but this time cite "data not published." I'm of the strong opinion important claims cannot be supported with "data not published." If it's true that this finding has never been published, then no mention of it should be made in the paper. If it has been published, then the citations should be corrected to refer to the correct paper.
Author Response
Point 1: For instance, line 349 cites reference #41 to refer to a study of the Arabidopsis SERK1 upstream regulatory sequence, but the citation is Becker et al 1992, describing binary vectors. Please check all citations to ensure they refer to the correct papers.

Response 1: You are right. We have revised “reference #41” to the correct one. Besides, we have also checked all citations to ensure they refer to the correct papers.
Reference #41:Valérie, H.; Jean-Philippe, V.; Marijke, V. H.; Ed, D.L.S.; Kim, B.; Ueli, G.; Sacco, C. V. The Arabidopsis Somatic Embryogenesis Receptor Kinase 1 Gene Is Expressed in Developing Ovules and Embryos and Enhances Embryogenic Competence in Culture. Plant physiology. 2001, 127(3), 803-816.
Point 2: A bigger issue is line 64 where the authors cite reference #14 to claim they have shown over-expressing SERK1 in pineapple doubles the efficiency of SE. However, that paper doesn't make that claim, rather it only describes the expression domain and dynamics of SERK1. Later (line 323) the authors again mention doubling SE in pineapple, but this time cite "data not published." I'm of the strong opinion important claims cannot be supported with "data not published." If it's true that this finding has never been published, then no mention of it should be made in the paper. If it has been published, then the citations should be corrected to refer to the correct paper.
Response 2: Sorry for the misleading. As our previous analysis, over-expressing SERK1 in pineapple doubles the efficiency of somatic embryogenesis. It is a part of the works in a master's thesis of Wentian Xu, a master of Horticulture Collage of South China Agricultural University. The title of the master thesis is “Cloning of AcSERK Promoters and Agrobacterium-mediated Transformation of Pineapple”. However, these data haven’t been published as a paper. Thus, we couldn’t cite it currently. Here, we append the related abstract in her master's thesis and wish that it could address your concerns.
The effects of AcSERK1 on somatic embryo induction were investigated, using callus from the positive plantlet transferred AcSERK1 as starting material. On average, each gram of non-embryogenic tissues could induce 9.22 embryogenic tissue particles by embryogenic induction about 40 days in darkness, higher than the control (74.0%); similarly, each gram of non-embryogenic tissues could induce 17.12 somatic embryos after differentiated cultivation about 30 days in light, and the somatic embryogenesis induction quantity was higher than 97.5%. The results showed that AcSERK1 might play a pivotal role in somatic embryo induction from non-embryogenic to embryogenic cells, and could enhance the somatic embryogenesis.

Round 2
Reviewer 1 Report
The authors have addressed all my points satisfactorily.